# Self-Monitoring Diabetes-Related Foot Ulcers with the MyFootCare App: A Mixed Methods Study

**DOI:** 10.3390/s23052547

**Published:** 2023-02-24

**Authors:** Bernd Ploderer, Damien Clark, Ross Brown, Joel Harman, Peter A. Lazzarini, Jaap J. Van Netten

**Affiliations:** 1School of Computer Science, Queensland University of Technology, Brisbane, QLD 4000, Australia; 2Metro North Hospital and Health Service, Herston, QLD 4029, Australia; 3School of Public Health and Social Work, Queensland University of Technology, Brisbane, QLD 4000, Australia; 4Amsterdam UMC, Department of Rehabilitation, University of Amsterdam, 1105 AZ Amsterdam, The Netherlands; 5Amsterdam Movement Sciences, Program Rehabilitation and Development, 1105 AZ Amsterdam, The Netherlands

**Keywords:** mobile health, patient generated health data, medical selfie, augmented reality, foot ulcer, diabetic, self-care (rehabilitation), therapeutic adherence and compliance, patient engagement, podiatry

## Abstract

People with diabetes-related foot ulcers (DFUs) need to perform self-care consistently over many months to promote healing and to mitigate risks of hospitalisation and amputation. However, during that time, improvement in their DFU can be hard to detect. Hence, there is a need for an accessible method to self-monitor DFUs at home. We developed a new mobile phone app, “MyFootCare”, to self-monitor DFU healing progression from photos of the foot. The aim of this study is to evaluate the engagement and perceived value of MyFootCare for people with a plantar DFU over 3 months’ duration. Data are collected through app log data and semi-structured interviews (weeks 0, 3, and 12) and analysed through descriptive statistics and thematic analysis. Ten out of 12 participants perceive MyFootCare as valuable to monitor progress and to reflect on events that affected self-care, and seven participants see it as potentially valuable to enhance consultations. Three app engagement patterns emerge: continuous, temporary, and failed engagement. These patterns highlight enablers for self-monitoring (such as having MyFootCare installed on the participant’s phone) and barriers (such as usability issues and lack of healing progress). We conclude that while many people with DFUs perceive app-based self-monitoring as valuable, actual engagement can be achieved for some but not for all people because of various facilitators and barriers. Further research should target improving usability, accuracy and sharing with healthcare professionals and test clinical outcomes when using the app.

## 1. Introduction

Diabetes-related foot ulcers (DFUs) are a leading cause of morbidity, mortality, and healthcare cost burdens globally [1,2]. In Australia alone each day, 50,000 people live with a DFU, 1000 are in hospital, 12 have an amputation, and 4 die because of a DFU, at an estimated annual direct cost of AUD 1.6 billion [3,4]. Best-practice treatment of people with DFU requires multidisciplinary team treatment in specialised DFU clinics, with various clinicians working together typically (bi-)weekly over several months to provide effective clinical care to heal the DFU [5]. However, the majority of DFU care performed is by the patients themselves or their carers away from the clinic, as self-care. Recommended DFU self-care typically includes patients regularly changing wound dressings, checking their DFU for changes and infection, and wearing offloading devices to relieve pressure and protect the ulcer [5]. Such recommendations are typically implemented in consultation between patients, carers, and clinicians to fit within the personal circumstances of the patient.

Self-monitoring is a key component of self-care [6]. For people with DFUs, self-monitoring holds potential to offer awareness of DFU healing and the impact of daily behaviours and self-care. Self-monitoring can also provide information to recognise complications and mitigate risks such as hospitalisation and amputation. This is important because patients often have a limited understanding of DFUs and the significance of self-care on DFU healing [7]. Poor mobility combined with the location of their DFU (most often on the plantar surface of their foot) also means that patients can find it difficult to see, reach, and care for the DFU themselves [8]. Perhaps most importantly, patients need to be able to perform self-care consistently over many months of DFU treatment [9], and during that time DFU healing changes can be hard to detect on a daily basis, which can be demoralising [8]. Hence, experts recommend that there is a need for simple and accessible methods for patients and carers to monitor DFUs at home [10].

To address this need, we created a new mobile phone app, “MyFootCare”, for patients and carers to monitor DFUs in their own homes and to receive encouraging feedback. MyFootCare encourages people to use their smartphone camera to take a digital photo of their foot. It uses visual analytics to help patients extract and monitor DFU size from their foot photo to track their healing progress more objectively. Similar mobile apps for monitoring DFU healing have been developed for use in healthcare services [11,12], with clinicians stating that such an app could potentially aid people with diabetes in checking their feet at home [11]. A study based on an imagined app showed that DFU patients also see potential in using photos for DFU self-monitoring [13]. Several systems have been developed to monitor feet at home [12,14,15,16], but these systems were largely used by people at risk of developing DFU, who had their photos assessed remotely by healthcare professionals. Contrary to these, MyFootCare was designed for people who already have a DFU to help them see, monitor, and care for a plantar DFU.

The specific aims of this mixed methods, but predominantly qualitative, study were to investigate (1) the value of MyFootCare perceived by patients and (2) the enablers and barriers for engaging with MyFootCare in naturalistic conditions over extended periods of time.

## 2. Materials and Methods

### 2.1. MyFootCare Design

MyFootCare was created by the investigator team through a user-centred design process with patients to identify self-care challenges [8] and to trial app prototypes [17,18], which have been described in detail elsewhere. In brief, three workshops with 14 podiatrists were conducted to further develop design ideas, conduct usability tests, and to obtain their support for this study.

MyFootCare was implemented as a fully functional Android app, based on Java frameworks and OpenCV [19], a free real-time computer vision development library. To measure the DFU size from foot images, the app uses a morphological watershed algorithm [20] provided by OpenCV to segment the foot from the image background and then the ulcer from the foot. The app relies on the surface area of the foot as a scale to measure DFU size. This approach was taken for usability reasons, so that users did not need to provide a reference point (like a ruler or a sticker on the foot) when taking a foot photo. As a result, the ulcer size was not measured as an exact cm^2^ measurement but as a percentage of the ulcer size from the first foot photo taken. The mobile phone flash was used to evenly illuminate the foot and keep the background dark. The prototype was developed and evaluated on a Samsung Galaxy S8 mobile phone.

MyFootCare offers the following features:**Progress graph**: users can monitor DFU size over time through a graph on the home screen (Figure 1a). The graph starts at 100% based on the DFU size from the first photo taken. A star on the graph visualises the goal to reach a 50% reduction within 4 weeks (Figure 1b). Reaching this goal can predict complete wound healing over an extended 12-week period [21].**Foot check**: to monitor the DFU size, users need to have a photo taken of their foot by another person, such as a relative or carer (Figure 2). Next, users need to manually analyse the photo by drawing lines to assist the app in segmenting the ulcer from the foot and the background (Figure 3). Users can review the foot image to identify changes and complications (e.g., infection) and add notes for discussion with their podiatrist or general practitioner. Finally, users receive feedback on their ulcer size, quantified as a percentage of the ulcer size from the first foot check (Figure 3d). Feedback messages and badges are customised to provide encouraging feedback tailored to their progress (e.g., “You’ve Reached Your Goal!”; see Appendix B for all messages).**Photo gallery**: users can review all their foot checks (image, progress, and notes) and show them to healthcare professionals via their gallery (Figure 1c).**Motivational image**: the upper half of the home screen shows an image that visualises a goal that users wish to achieve when their DFU has healed (e.g., be able to walk the dog, Figure 1a). Users could choose from 10 different images or upload their own photo. This is important because setting a realistic goal based on something that people want to achieve (rather than avoid) is typically one of the first steps in establishing a self-care plan [22,23].**Notifications**: users receive reminders to take a foot selfie. The timing and message of reminders can be tailored to fit with the time of their dressing changes (Figure 1d).

### 2.2. Trial Study

#### 2.2.1. Study Design

The study was designed as a mixed methods prospective 3-month cohort study, as this period is the typical DFU healing duration and recommended by international criteria for clinical studies [24]. The study was predominantly qualitative, based on semi-structured interviews. Quantitative measures were created from app log data and ratings of app features. Ethics approval was obtained from the Prince Charles Hospital (#HREC/18/QPCH/185), and the study was registered before commencement (ACTRN12618001276246).

#### 2.2.2. Participant Recruitment

Eligible participants were people with a DFU on the plantar surface of the foot who were treated at a diabetic foot clinic, who owned a smartphone and who were assisted in their DFU management by a carer (e.g., a spouse, or home care nurse). DFUs were defined as a full thickness wound on the foot (i.e., below the malleoli) of a person with diagnosed type 1 or type 2 diabetes mellitus [1]. A smartphone was defined as an internet-enabled mobile phone and was a requirement so participants would have some familiarity with mobile applications.

Participants were recruited through three clinics in Brisbane, Australia. Eligible patient participants and their carers were invited to participate by their treating clinician. As a further incentive, participants were offered an AUD 50 voucher at the start of the study and another AUD 50 voucher if they completed the study. During the entire study period, participants continued to receive standard care from their clinic.

The recruitment target of 12 patients was based on our prior experience with reaching data saturation in similar studies. Prior research also shows that data saturation often occurs after 12 interviewees [25]. A total of 12 participants is also the most common sample size in human–computer interaction studies as it is seen to provide a balance between cost and return on significant insights on user engagement digital technology [26].

#### 2.2.3. Data Collection

Demographics and DFU information were recorded by their podiatrist in the validated Queensland High Risk Foot Form clinical record [27].

Semi-structured interviews were conducted at weeks 0, 3, and 12 in person at the participant’s clinic by one of the investigators (DC, BP) based on an interview guide (Appendix A). Interviews were audio-recorded and transcribed verbatim. Interactions with MyFootCare during the interviews were screen-recorded.

Before interview 1, the investigators used MyFootCare to take a high-quality foot photo during the participant’s consultation. This photo served as a baseline (100%) for further foot checks.

Interview 1 (day 0, 60 min) focused on understanding the participant’s background and on introducing MyFootCare. First, we discussed the participant’s DFU and diabetes history, self-care practices at home and their impact on their everyday life, and their smartphone use. Second, we showed participants all features of the MyFootCare app and we used a think-aloud technique [28] to gain their initial impressions and questions about each app feature. Finally, participants rated the perceived usefulness of each app feature on a scale from 1 (not useful) to 10 (very useful) and explained their rating.

During interview 1, participants either received a smartphone (Samsung S8) for the study duration with MyFootCare installed on it, or we installed MyFootCare on their personal phones if the participants preferred and it was a similar phone model. Participants were asked to use MyFootCare each time they changed their wound dressing away from the clinic. They also received a stylus and a printed guide to assist with their foot checks.

After 10 days, participants received a phone call from one of the investigators to check if they had questions or needed technical assistance.

Interview 2 (week 3, 45 min) focused on understanding the participant’s initial app engagement. First, participants were asked to reflect on their progress. Second, participants were asked to open their MyFootCare app and show the investigators how they used the app and to raise any usability issues they may have experienced. Finally, participants rated the perceived usefulness of each app feature using the same scale as in interview 1, having now used the app for several weeks.

Interview 3 (week 12, 60 min) focused on understanding the participant’s ongoing app engagement and perceived value of MyFootCare for self-care. First, participants were asked to reflect on the progress of their DFU and self-care. Second, participants were asked to open their MyFootCare app and tell the investigators about their app data (graph and photo gallery), their experiences with taking photos and analysing them, and their reflections on factors that may have supported or impeded their progress and app use. Finally, participants rated the perceived usefulness of each app feature again and reflected on the value provided.

Quantitative data on MyFootCare engagement were automatically collected on the participant’s phone by the app. MyFootCare automatically logged each interaction with the app with timestamps in a log file and all foot images taken were stored on the phone. A second log file contained the time stamp and wound size for each foot check. Log files were exported at the end of interview 3.

#### 2.2.4. Data Analysis

We analysed the data based on the principles of a reflexive thematic analysis approach [29,30]. In contrast to positivist thematic analysis, a reflexive approach highlights that themes are not discovered but generated by the researcher [30]. The researcher plays an active role in the thematic analysis, and their interpretation of the data and the themes generated are shaped by their background and by theoretical frameworks [29]. In this project, the analysis was led by the first two authors—a human–computer interaction researcher (BP) and a podiatrist (DC). The following frameworks guided our analysis:Self-care, which the WHO defines as “the ability of individuals, families, and communities to promote, maintain health, prevent disease, and to cope with illness with or without the support of a health-care provider” [31].Perceived value of patient-generated health data [32,33] for patients, which is defined as functional (to support health outcomes), emotional (understand and regulate emotions), social (share experience and support with peers), transactional (enrich consultations with health professionals), efficiency (eliminate unnecessary appointments), and self-determination value (empower patients).User engagement, which is defined as the process of how people start and continue to use technology for a certain purpose [34], which can be cyclical and include phases of disengagement and re-engagement [35]. Engagement is different from adherence, in that it is more dynamic and shaped by subjective factors such as a person’s goal, challenge, and experience with technology [36].

Our reflexive thematic analysis consisted of:Familiarisation: all interviews were transcribed verbatim by a professional transcription service. Immediately after each interview, we created notes with initial observations about the value and engagement with MyFootCare.Generating initial codes: coding started after completing the interviews. We created inductive codes both deductively (e.g., perceived functional value) and inductively to create more specific codes for our particular context (e.g., functional value from seeing the plantar side of the foot). The qualitative data analysis software NVivo was used to code all transcripts.Constructing themes: our research aims were used to construct themes that provide a coherent and insightful account of how patients engage and the value they perceive. Crosstab queries in NVivo allowed us to compare the frequency of codes between participants with different engagement levels.Reviewing and defining themes and producing the report: these three phases were intertwined and iterative. Themes were reviewed through regular meetings with all authors. Definitions were created and refined throughout the writing of the report and reviewed with all authors until consensus was reached.

All quantitative data were analysed using Microsoft Excel for Mac 16. Descriptive statistics used to display variables included frequencies (proportions), mean (standard deviation (SD)), and median (interquartile range (IQR)).

## 3. Results

### 3.1. Participants

All participant characteristics are displayed in Table 1. In total, we recruited 12 participants; 9 (75%) were male, median (IQR) age 53 (46–65) years, and 10 (83%) had type 2 diabetes. Three (25%) participants had MyFootCare installed on their personal Android phone and nine (75%) received a Samsung S8 smartphone with MyFootCare installed for the study duration. All 12 participants completed the study with a third and final interview at week 12. Only P02 asked to conduct his final interview earlier at week 7 because of a health concern.

### 3.2. Value from Digital DFU Self-Care

Table 2 shows participants’ ratings of the perceived usefulness of MyFootCare at weeks 0, 3, and 12. The median (IQR) rating of the overall MyFootCare app was 10 (10, 10) (“very useful”) from a 10-point Likert scale at week 0, and this did not change at week 3 and week 12. Of the individual app features, participants also rated as very useful the ability to track progress through a graph (10 (8,10)), see wound photos in the gallery (10 (10,10)), and share data with health professionals (10 (3,10)) throughout the study, whereas participants rated the reminder notifications (8 (5,10)) and motivational image (6 (4,9)) features lower and these scores decreased throughout the study.

The qualitative data presented below provides the reasons behind the ratings and the value that participants perceived (based on [32]). Participants with limited app engagement sometimes rated the app’s potential rather than the value that materialised for them.

#### 3.2.1. Functional Value: Monitoring Progress through Graph and Photo Gallery Features

Performing digital foot checks with MyFootCare provided clear value to DFU care. Ten participants reported that MyFootCare supported their health outcomes (functional value) because it allowed them to see the progress of their ulcer with their own eyes.


*“I think it’s a necessity. If you involve the patient, like I was involved with it, so of course I was keen to see it, and you’re obviously keen to see it go down.”*
(P09)


*“I think that’s fantastic. Yes, I’d give that a 10, because you know what’s going on. Because how are you to know if you don’t see it?”*
(P08)

#### 3.2.2. Functional Value: Reflection on Care and Health Decisions

Participants valued MyFootCare because it allowed them to reflect on events that affected their care and to aid with health decisions. During interviews, all participants used the graph to reflect on the reasons for changes in their wound size, for example, because they were more active on their feet. Five participants reported that reflection on MyFootCare data played a role in care decisions.


*“I did write something on it [added a note to the photo]. Tinge of green. Well, the smell. The sign of infection.”*
(P10)


*“It aided in the decision to stop work and improve.”*
(P02)

#### 3.2.3. Transactional Value: Enhancing Consultations with Healthcare Professionals

The participants saw clear potential value in sharing MyFootCare data with a health professional to aid with clinical consultations. Seven participants reported that they intend to show MyFootCare to their health professional (GP, endocrinologist, and surgeon) to review the ulcer progress without having to take off and re-dress the bandage. These health professionals lacked equipment to re-dress their ulcers, and hence participants were reluctant to take off their dressing during consultations. Three participants also wished to share MyFootCare data electronically, e.g., by sending an email.


*“But yeah, I think it’s a good idea, because when I do go to my GP, I can show her the photos and how I’ve progressed, and now it’s healed.”*
(P03)


*“If people think that there might be an infection, instead of ringing up and making an appointment maybe send a photo through and the nurse or the doctor could have a look at that and be like, ‘Oh, yeah, maybe you should come in,’ or at least put their mind at ease and say, ‘No, that can wait for our appointment.’”*
(P04)

In practice, participants often did not share their photos because healthcare professionals appeared busy and did not prompt patients to show photos or because they were taking their own photos.


*“From what I’ve seen with the GPs they seem to rush through. So I don’t know whether they would sit down and talk about it.”*
(P02)


*“They [podiatrists] took their own photos on the camera that they provided. They are not really that interested. Yeah.”*
(P01)

It is important to note that the value for consultations is not purely hypothetical because three participants shared foot photos with a healthcare professional during the trial study.


*“We’ll be going to our diabetic man this week and he’ll be interested to know how [P10]’s feet are going, and he used to say to us, can I have a look, and we’d say, no—because he has absolutely nothing there that he could possibly put it—bandage it up or anything.”*
(Carer of P10)

### 3.3. Engagement with MyFootCare

Figure 4 displays the log data over the 12-week duration of the study. Overall, participants used MyFootCare a mean (SD) 16 (11.5) times over the 12-week duration or 1.4 (1) times per week. Three different usage sub-groups emerged with four (33%) participants “continually” using (28 (11.3) over 12 weeks; ≥20 digital foot checks overall), four (33%) others “temporarily” using (16 (2.5) over 12 weeks; ≥10 and <19 foot checks), and the other four (33%) “failing” to use MyFootCare after the first few weeks (5 (1.4) over 12 weeks; <10 digital foot checks).

Our qualitative findings seemed to confirm and further tease out the three distinct patterns identified in the quantitative findings. Enablers and barriers to engagement for each group are described below. The engagement level of each participant is indicated through a “C”, “T”, or “F”, e.g., “P03C” for continuous engagement of participant 3.

#### 3.3.1. Enablers for Continuous Engagement

Participants who continuously engaged with MyFootCare were also dedicated to diabetes and DFU self-care. The following enablers allowed them to integrate MyFootCare with their self-care routines, although we noted some enablers were also identified by other participants.

**MyFootCare installed on personal phone:** all three participants who had MyFootCare installed on their personal phone were continuous users. This enabled them, as they had MyFootCare with them all the time. For example, P03C even took photos during her stay in hospital, which also meant she could show photos to the surgeon. It also meant that other features, such as reminder notifications and the motivational image, were more accessible.


*“I think it’s very useful, because you can show your doctor or the surgeon the progress of what’s been going on with your foot.”*
(P03C)

**Familiarity with foot selfies**: six participants already had photos of their own foot ulcer on their personal phone. They were used to reviewing these photos and to reflecting on the reasons that promoted or inhibited healing. However, prior to using MyFootCare, foot photos were usually mixed in with all their personal photos and difficult to retrieve.


*“I have all of them [on the phone]. They’re taken for other medical viewing by doctors, and whatever.”*
(P10C)

**Dedicated caregivers who take high-quality photos:** it took time for carers to learn how to take a good photo and to understand how this impacts the accuracy of the analysis. We saw that all four continuous users had caregivers who put in considerable effort to ensure they produced high-quality photos. For example:


*“See, the boys got down on their knees and took it straight on, which you need to do to get that exact outline.”*
(P03C)

#### 3.3.2. Barriers Leading to Temporary Disengagement

Four participants engaged with MyFootCare for parts of the trial study (Figure 4). The following barriers were not directly related to MyFootCare, yet they were stated as the main reasons for pausing their engagement.

**Work commitments**: a main barrier was the tension between (paid and unpaid) work commitments and the need to care for and rest the foot. For example, one participant described the tension between having to work to support his family and needing to rest to heal the ulcer as a “Catch-22”.


*“I’m in a real Catch-22 now. I know health’s more important, but so is your family. So it’s a really sticky situation. I mean, I’m not a rich person. Obviously, if I was, I’d take the time off and sit at my house for three months and not do anything. Which is probably what I need to do. But I just can’t afford to do that.”*
(P11T)

**Health disruptions**: a second barrier for engagement were health disruptions. For example, one participant permanently disengaged from MyFootCare when he was hospitalised because of complications arising from cancer, followed by almost daily health appointments. The MyFootCare app was secondary to these needs during that time.


*“I spent nine days in hospital and quite a bit of other stuff. And then like with the prostate cancer and I got—every day of the week we were at the doctors giving blood, at the hospital, podiatrist. It’s just been full on, absolutely full on.”*
(P07T)

#### 3.3.3. Barriers Leading to Failed Engagement

Four participants failed to engage with MyFootCare: they used the app occasionally at the start of the study but generally stopped using MyFootCare after a few weeks (Figure 4). These participants highlighted barriers related to their health, as well as barriers related to the usability of MyFootCare.

**Frustration with lack of healing progress:** three participants reported frustration with the lack of progress on their ulcer, which also affected their engagement with MyFootCare as they could not see any progress on the graph.


*“To be honest, I’m at the point where I just want my foot cut off. It has made a huge impact on my life.”*
(P04F)

**Lack of confidence using smartphones**: during the interviews, we observed that the skills and confidence with which participants used smartphones varied considerably. Participants with lower confidence reported using their own phone primarily for communication and rarely for apps or to take photos. We observed that some participants navigated MyFootCare with difficulty because they were not used to the shape and operating system of the study phone, or they found it difficult to read information.


*“Without my glasses I can see the graph, but I can’t really see the numbers.”*
(P04F)

**Frustration with having to re-do foot checks:** having to retake the photo or repeat the analysis was a major source of frustration, which we also witnessed during some of our interviews when we analysed photos together with participants.


*“The greatest bother was if I put my finger in one spot and started the outline, and I noticed it was a bit close, then having to go back and then having to do all of the outlines again. … To me, it was kind of annoying, sorry to say.”*
(P04F)

The main reason for having to redo foot checks were poorly taken photos. This was the case when photos were taken from above rather than parallel to the foot, when they did not show the whole foot, or when they contained background distractions like bright lights or the skin of the leg and arm (Figure 5a). A second reason was that the lines drawn during the analysis to separate the background, foot, and ulcer were poorly drawn. Instead of drawing around the foot and staying away from the edge, we observed how participants tried to cut as close to the edge as possible, similar to cutting out a picture. Some participants drew too many lines and thereby cut across edges, like when trying to colour in an area (see Figure 5b). Participants also struggled with the last step of the analysis, where they needed to tap the wound without touching the edge or outside of the wound. We provided a stylus to assist them with this step. However, images with small wounds or wounds close to the edge of the foot remained a challenge.

**Lack of accuracy and reliability:** inaccurate foot check results and limited reliability between different foot checks performed by the same participant were also a major barrier. This was largely a consequence of the limitations of MyFootCare (limited camera resolution and non-standardised photos) and of problems associated with the manual analysis process (when participants had enough of redoing foot checks and accepted inaccurate outlines of the foot and wound, as described above). In addition, occasional analysis errors made the graph difficult to read because these outliers changed the scale of the graph and minimised the actual progress (see Figure 5c).


*“Yeah, I don’t know, sort of, because it’s saying, like, 800–8000 percent bigger and, and 2000-something percent bigger. It’s just—you probably see—it doesn’t really do jack shit, in other words.”*
(P01F)

## 4. Discussion

### 4.1. Principal Results

The majority of participants perceived self-monitoring DFUs with MyFootCare as valuable for their self-care, in particular to see ulcer healing progress from the foot check feature (photos and graph). Similar to previous studies [14,15,16], we found that the majority of people perceived foot photos as valuable for self-monitoring at home. Additionally, MyFootCare provided a progress graph to provide objective DFU size information. While such a feature has been mentioned as potentially useful in previous studies by clinicians [11] and by patients imagining such an app [13], the current study presents experiences from patients actually using progress data in their daily lives. The results showed that most patients valued such data to verify subjective observations from the photos, even though they recognised that the accuracy of the progress graph was limited. Furthermore, the timeline of the graph encouraged many participants to reflect on actions and events that affected their self-care and healing progress. Such personal reflection is important because it can lead to patients feeling a higher degree of control in their health care, congruent with psychological empowerment [37].

Participants saw potential value in sharing MyFootCare data with healthcare professionals who typically do not treat or view the ulcer during consultations, such as their general practitioner. Thus, capturing photos can help patients to prepare for consultations and take on a more active role in their interactions with healthcare professionals, as found in other studies, e.g., by recalling information and health decisions [38,39]. Reviewing photos during consultation also benefits healthcare professionals because it prompts discussion about health experiences [40], adherence to treatment plans [41], and the broader lives of patients [42]. Unfortunately, such sharing rarely occurred in this study because participants perceived healthcare professionals as too busy and not interested in their observations. This observation also aligns with recent studies that show that patients rarely share their data unless they get asked by the clinician [43], and clinicians spend little to no time asking patients [44]. This seems a missed opportunity, and in Section 4.2, we recommend several practical implications that may improve this in future.

We identified and then explored three distinct engagement patterns with MyFootCare from participants: continuous, temporary, or failed engagement. Understanding these patterns is important because they reflect the reality of digital health interventions, where many systems are seen as valuable but have mixed uptake because of challenges with accessibility, privacy, and accuracy [33]. The patterns in this study were comparable to similar studies of digital systems for DFU self-care. An 8-week trial by Anthony et al. [16] showed that 77% of users were willing to take regular foot photos once a week, which is similar to the 66% (continuous or temporary) users in our study who took on average at least one photo per week. A 6-month trial of the “Foot Selfie” system [14] showed that 93% of participants were imaging their feet at least every other day. However, in both studies, photos were taken primarily to screen the foot for ulcers. In contrast, the participants in our study used MyFootCare as part of their self-care routine when they changed wound dressings. They used MyFootCare more continuously when it was on their own phone, when they were familiar with foot selfies, and when they had a dedicated carer to take high-quality photos.

In examining the barriers to engagement, we found that participants took breaks from using MyFootCare when they experienced health disruptions, e.g., when they received care in hospitals. As predicted by technology engagement frameworks [34,35], we found that participants re-engaged when they were back at home.

Participants stopped using MyFootCare when they had their DFU for a long time and could not see any progress. Our findings align with related work with chronic disease patients [6], which suggests that their willingness to self-monitor is associated with the sense of control that they perceive over the disease. If patients feel that they cannot control health outcomes and do not expect to see any improvements, then they are less willing to self-monitor because the cost (in terms of time and effort required) is perceived to be higher than the anticipated benefits [6]. Participants also stopped using MyFootCare permanently when they did not feel confident using smartphones or when they experienced frustration from ongoing usability problems. These barriers could potentially be addressed by caregivers who are comfortable using smartphones and who can conduct the manual analysis (as well as take photos). We seek to address usability barriers in future work (see Section 4.3), but we do not expect that this would change the low engagement levels of long-time DFU patients with MyFootCare. These patients require a different approach [5].

### 4.2. Practical Implications

For healthcare professionals interested in engaging DFU patients more actively in their self-care through digital systems or foot photos, we suggest three recommendations. First, it seems important to identify suitable patients who are more likely to engage. For example, patients are more likely to engage if they have suitable smartphones and feel confident using them. DFU patients are often older adults [45], and diabetes can also negatively affect their vision [46] and their hand function and dexterity [47], which can all affect their use of smartphones. Patients require the support of a caregiver to take high-quality foot photos in their own home. Familiarity with having foot photos taken is also beneficial.

Second, digital self-care worked best for patients and carers who were already dedicated to self-care and who were looking for additional ways to promote DFU healing. MyFootCare did not motivate participants who had difficulties with adhering to other forms of self-care, such as dressing changes and offloading. Such patients need other forms of (behavioural) support [8].

Finally, healthcare professionals need to provide ongoing support to patients. Initially, they need to help them set up MyFootCare with the first foot check. Follow-ups are required to assist patients with usability issues, to actively inquire during their consultation into data collected, and to promote the integration of digital foot checks with existing dressing-change routines. Like others [48], we suggest that healthcare professionals can benefit from training, so that they can educate their patients on digital technologies and elicit patient data during consultations more effectively.

### 4.3. Limitations and Future Work

This study was based on 12 participants, which limited the validity of the quantitative results presented. Whilst quantitative information is provided in figures and tables, this information is descriptive only, and the main findings about the perceived value of MyFootCare, as well as the barriers and enablers for engagement, are based on qualitative data. Due to the small cohort and the 3-month period, we did not find an association between app engagement and ulcer healing; however, our methods were also not designed to detect such an association. If we would have found one, it could also have been by chance. A larger and primarily quantitative study is needed to investigate such an association. Instead, the strength of the current study was its high ecological validity through qualitative results about app engagement in real-world contexts over a 3-month period. This 3-month period was also sufficient to achieve our aims to evaluate engagement and app usage, as app engagement did not change after week 6.

This study identified several barriers that need to be addressed in future work. First, more work is required to simplify the foot check. Related work [16] suggests that patients benefit from having a selfie stick to take their own photos. We had tried selfie sticks in our usability tests but found them too difficult to use as they required long arms and flexibility. However, we see potential in a dedicated apparatus such as the “Foot Selfie” system [14], which consists of a phone holder and an apparatus to rest their foot on and allows patients to take photos on their own with minimal training.

Second, patients would benefit from simplifying the analysis to segment ulcer and foot in an image and to improve the accuracy and reliability. The current analysis requires a manual process to identify the wound patient, which can help to engage the patient but which also introduces subjectivity to the segmentation of the DFU. Even with clinicians, studies document that manual wound measurements involve a trade-off between accuracy and feasibility (time required and risk of contamination) [49]. To address these limitations, we see large potential in machine learning techniques, where recent studies provide reasonable results in segmenting DFUs in foot images [50,51]. We envision that analysis of foot images could be automated through a system that securely exchanges foot images taken on a patient’s phone with an internet-based analysis service. Furthermore, machine learning techniques also show potential to identify complications like infection and ischemia [52] in foot images, which could be used to alert patients to seek treatment.

Third, the current MyFootCare design is limited to tracking a single wound on the plantar surface of the foot. We focused on plantar wounds because these are the most common and are not visible to the patient [53]. However, participants with DFUs at the edge of the plantar surface reported difficulty with their analysis, as did participants with very small wounds. A different app design is needed to better support these patients.

Finally, MyFootCare did not allow remote monitoring by healthcare professionals. Patients can benefit from two-way communication with healthcare professionals to reduce the number of visits to the clinic [13]. When designing MyFootCare, we decided not to include electronic data sharing for a number of practical reasons: potential privacy risks, difficulty with diagnosing ulcers from images alone [54], and potential interference with the care provided by podiatrists. On a more fundamental level, we felt that remote monitoring by healthcare professionals would disempower the patients because it could increase their reliance on clinical care instead of empowering patients in their own care. Hence, in the spirit of participatory healthcare [55], we focused on how MyFootCare can provide patients with new insights for their own care, which they can share during consultations to give them a voice in their conversations with healthcare professionals.

## 5. Conclusions

For people whose DFUs are healing and who can access MyFootCare on their own phone, using an app for self-monitoring their DFU provides value through new health insights and through reflection on the events that promote or hinder progress. Successful engagement depends on various facilitators and barriers and can be achieved for some but not for all people. More work is needed to improve MyFootCare to address usability issues and to enhance its accuracy through standardised photos.

## Figures and Tables

**Figure 1 sensors-23-02547-f001:**
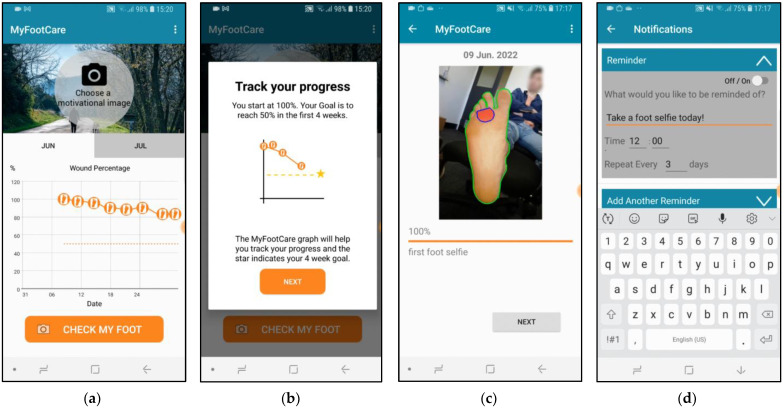
MyFootCare features. (**a**) A motivational image on the top of the home screen visualises a person’s goals (e.g., to go for a walk). A progress graph shows changes in DFU size over time. (**b**) A star on the graph visualises the goal to reach a 50% reduction within 4 weeks. (**c**) Patients can review all foot checks through a gallery. (**d**) Notifications to take foot selfies can be tailored under settings.

**Figure 2 sensors-23-02547-f002:**
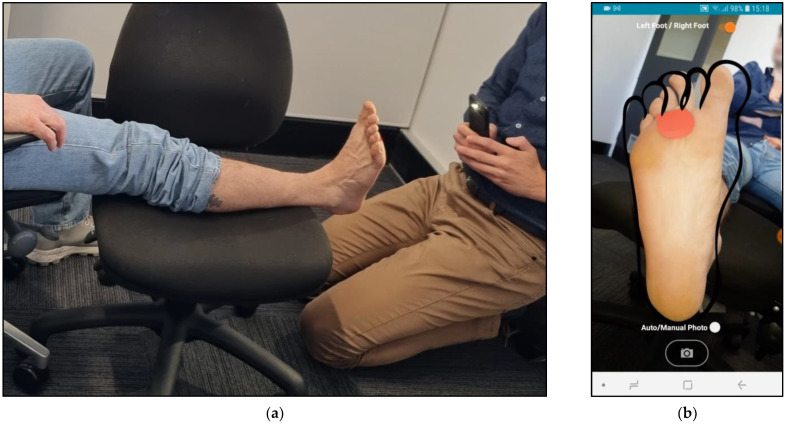
(**a**) Taking photos for a foot check requires the person with the DFU to rest their foot on a chair while another person takes a photo. (**b**) A silhouette supports photographers in taking photos at a consistent distance.

**Figure 3 sensors-23-02547-f003:**
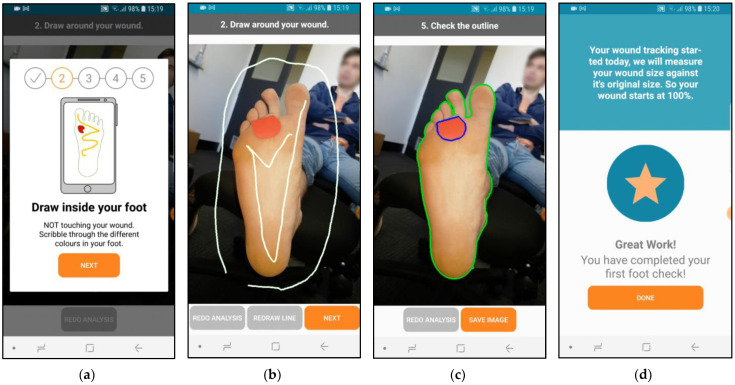
Foot checks involve a five-step process to segment the ulcer from the rest of the foot and the background: (**a**,**b**) users need to draw a line around the foot, draw inside the foot, zoom in on wound, draw inside the wound, and (**c**) check the outlines around the ulcer and around the foot. If the outlines are inaccurate, users can redraw the lines to get a more accurate result. (**d**) Users receive tailored feedback to encourage monitoring.

**Figure 4 sensors-23-02547-f004:**
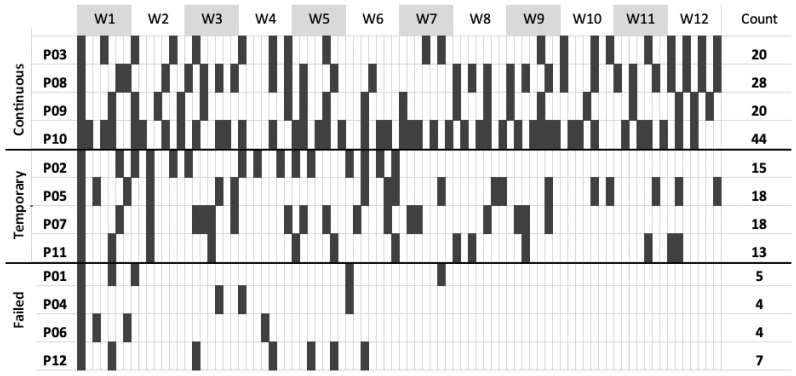
MyFootCare usage log data, showing the foot checks (photo and analysis) performed for each participant throughout the study period. Participants are clustered into three groups: continuous, temporary, and failed engagement.

**Figure 5 sensors-23-02547-f005:**
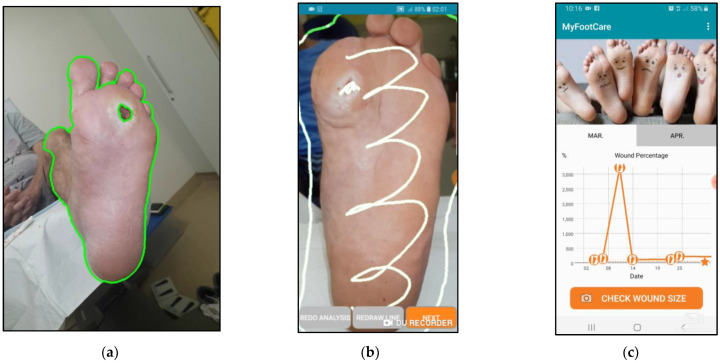
Examples of problems. (**a**) Background distractions such as skin of the leg and arm led to poor analysis results, with the green outline of the foot also covering the leg (P01F). (**b**) The white lines show the lines drawn by P12F during the analysis. The scribble drawn to indicate the ulcer cuts across the edge of the ulcer, and therefore, the analysis did not work. (**c**) Outliers in the data that were due to analysis problems changed the scale of the graph and made the progress difficult to see.

**Table 1 sensors-23-02547-t001:** Participant characteristics (number (%) or median (IQR), unless otherwise stated).

Characteristics	Total
Numbers	12
Age (years)	53 [46; 65]
Males	9 (75%)
Type 2 Diabetes	10 (83.3%)
Occupation	
Employed	3 (25%)
Unemployed	4 (33.3%)
Retired	5 (41.7%)
Smartphone owned	
Android	9 (75%)
iOS	3 (25%)
MyFootCare access on	
Study phone	9 (75%)
Personal phone	3 (25%)
Ulcer duration history (months)	6 [5; 12]
Ulcer location	
Digit/toe	2 (16.7%)
Forefoot	8 (66.7%)
Midfoot	1 (8.3%)
Heel	1 (8.3%)
Ulcer healing status at week 12	
Healed	1 (8.3%)
Decreased in size	5 (41.7%)
Increased in size	6 (50%)

**Table 2 sensors-23-02547-t002:** Participant ratings of MyFootCare app usefulness, reported as median (IQR).

App Feature *	Week 0 (n = 12)	Week 3 (n = 12)	Week 12 (n = 11 ^)
MyFootCare app overall	10 [10; 10]	10 [7; 10]	10 [8; 10]
Motivational image	6 [4; 9]	6.5 [3; 10]	5 [3; 9]
Notifications	8 [5; 10]	6 [4; 10]	3 [2; 9]
Progress graph	10 [8; 10]	10 [9; 10]	8 [8; 10]
Photo gallery	10 [10; 10]	10 [8; 10]	10 [10; 10]
Share data with a health professional	10 [9; 10]	9.5 [9; 10]	10 [8; 10]

* Reponses to questions/items of ‘how useful are the different app features/app as a whole for you?’ (1 = not useful; 10 very useful). ^ Only 11 participants reported, as 1 participant reported open-ended feedback without a rating.

## Data Availability

Not applicable.

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
