# Peer review of "Self-Monitoring Diabetes-Related Foot Ulcers with the MyFootCare App: A Mixed Methods Study"

_sensors, 2023, doi:10.3390/s23052547_

Round 1
Reviewer 1 Report
This is a very interesting manuscript and I would like to thank you for the opportunity to review.
A few questions/suggestions:
1. Since DFU takes up to many months to heal, why did you choose a 3-month period for your study? Do you think that choosing different periods or time would affect the patients' compliance with the app using regime?
2. Is there an online material or an appendix to your manuscript showing a sample of the semi-structured interview?
3. Please include in the discussion section a short mention of how the caretaker (family or other) could alleviate some of the limitations of the app usage.
4. In your keywords you should include "augmented reality". Although your app is not exactly AI, including the term in your keywords will ease people finding your manuscript during their literature search.
Author Response
Dear editor,
We would like to thank you and the reviewers for your assessment of our Sensors Special Issue submission on DFU self-monitoring with the MyFootCare app. We were pleased to read the positive remarks from the editor and both reviewers, have taken their comments on board and have revised our manuscript accordingly. We believe that this has significantly improved the manuscript. Below we provide a point-by-point response to comments and we have tracked changes to the manuscript text. Responses are separated from the comments. Manuscript text is added in italics in this point-by-point response, with revisions underlined. Line numbers are based on the manuscript without markups shown.
We hope the editor and reviewers enjoy reading this new version of our manuscript, and we look forward to their assessment.
Yours sincerely, on behalf of all authors,
Bernd Ploderer and Jaap van Netten
Reviewer 1
COMMENT 1:
This is a very interesting manuscript and I would like to thank you for the opportunity to review.
A few questions/suggestions:
Since DFU takes up to many months to heal, why did you choose a 3-month period for your study? Do you think that choosing different periods or time would affect the patients' compliance with the app using regime?
RESPONSE:
We thank the reviewer for stressing this important aspect of our methodology. We agree that the period is an important consideration which is likely to impact app engagement (and also participant recruitment).
As outlined in section 2.2.1, we chose a 3-month period because this is the typical DFU healing duration and recommended by international criteria [9]. As described in Section 4.1, para 2, this is shorter than the 6-month study by Swerdlow et al. [6] and longer than the 8-week study by Anthony et al. [4].
Lines 135-137: prospective 3-month cohort study, as this period is the typical DFU healing duration and recommended by international criteria for clinical studies [9].
Lines 461-468: An 8-week trial by Anthony et al. [4] showed that 77% of users were willing to take regular foot photos once a week, which is similar to the 66% (continuous or temporary) users in our study who took on average at least 1 photo per week. A six-month trial of the “Foot Selfie” system [6] showed that 93% of participants were imaging their feet at least every other day.
Looking back at our findings, this period was too short to achieve healing for most participants (only 1 participant healed). However, it was more than long enough to achieve our aims. In hindsight, 6 weeks would have been sufficient for our aims, as interview 2 (week 3) provided reliable insights into perceived value, enablers and barriers, and because app engagement (adherence) did not change after week 6 (as shown in Figure 4). We have added this to the discussion.
Lines 521-523: This 3-month period was also sufficient to achieve our aims to evaluate engagement and app usage, as app engagement did not change after week 6.
COMMENT 2:
Is there an online material or an appendix to your manuscript showing a sample of the semi-structured interview?
RESPONSE:
Please find attached the interview guide for all three interviews, which can be made available as supplementary material.
Lines 163-165: Semi-structured interviews were conducted at weeks 0, 3 and 12 in person at the participant’s clinic by one of the investigators (DC, BP) based on an interview guide (Supplementary Material).
COMMENT 3:
Please include in the discussion section a short mention of how the caretaker (family or other) could alleviate some of the limitations of the app usage.
RESPONSE:
Good point. The last paragraph in Section 4.1 now includes a statement that caregivers may be able to help address barriers such as lack of confidence with smartphones and usability problems with the manual analysis of photos.
Lines 483-485: These barriers could potentially be addressed by caregivers who are comfortable using smartphones and who can conduct the manual analysis (as well as take photos).
COMMENT 4:
In your keywords you should include "augmented reality". Although your app is not exactly AI, including the term in your keywords will ease people finding your manuscript during their literature search.
RESPONSE:
We thank the reviewer for this thought. We have now added augmented reality to the keywords (line 29). Whilst this was not how we initially thought about MyFootCare, it could be seen as augmenting reality by overlaying virtual information on the foot image.
References
- Cassidy, B.; Reeves, N.D.; Pappachan, J.M.; Ahmad, N.; Haycocks, S.; Gillespie, D.; Yap, M.H. A cloud-based deep learning framework for remote detection of diabetic foot ulcers. IEEE Pervasive Computing 2022, 21, 78-86, doi:10.1109/MPRV.2021.3135686.
- Matilla, A.C.; Mena, A.C.; Giménez, A.A.; Fernández Morata, J. Role of the Healico© Wound Care Smartphone Application in Preventing a Foot Amputation in a 65-Year-Old Patient with Diabetes. Am J Case Rep 2022, 23, e936359, doi:10.12659/AJCR.936359.
- Boodoo, C.; Perry, A.J.; Hunter, J.P.; Duta, I.D.; Newhook, P.S.C.; Leung, G.; Cross, K. Views of Patients on Using mHealth to Monitor and Prevent Diabetic Foot Ulcers: Qualitative Study. JMIR Diabetes 2017, 2, e22, doi:10.2196/diabetes.8505.
- Anthony, C.A.; Femino, J.E.; Miller, A.C.; Polgreen, L.A.; Rojas, E.O.; Francis, S.L.; Segre, A.M.; Polgreen, P.M. Diabetic Foot Surveillance Using Mobile Phones and Automated Software Messaging, a Randomized Observational Trial. Iowa Orthop J 2020, 40, 35-42.
- Kilic, M.; KaradaÄŸ, A. Developing and Evaluating a Mobile Foot Care Application for Persons With Diabetes Mellitus: A Randomized Pilot Study. Wound management & prevention 2020, 66, 29-40, doi:10.25270/wmp.2020.10.2940.
- Swerdlow, M.; Shin, L.; D’Huyvetter, K.; Mack, W.J.; Armstrong, D.G. Initial Clinical Experience With a Simple, Home System for Early Detection and Monitoring of Diabetic Foot Ulcers: The Foot Selfie. Journal of Diabetes Science and Technology 2021, 0, 19322968211053348, doi:10.1177/19322968211053348.
- Goyal, M.; Reeves, N.D.; Rajbhandari, S.; Ahmad, N.; Wang, C.; Yap, M.H. Recognition of ischaemia and infection in diabetic foot ulcers: Dataset and techniques. Computers in Biology and Medicine 2020, 117, 103616, doi:10.1016/j.compbiomed.2020.103616.
- Jørgensen, L.B.; Sørensen, J.A.; Jemec, G.B.; Yderstræde, K.B. Methods to assess area and volume of wounds – a systematic review. International Wound Journal 2016, 13, 540-553, doi:10.1111/iwj.12472.
- Jeffcoate, W.J.; Bus, S.A.; Game, F.L.; Hinchliffe, R.J.; Price, P.E.; Schaper, N.C. Reporting standards of studies and papers on the prevention and management of foot ulcers in diabetes: required details and markers of good quality. The Lancet Diabetes & Endocrinology 2016, 4, 781-788, doi:10.1016/S2213-8587(16)30012-2.
- Guest, G.; Bunce, A.; Johnson, L. How Many Interviews Are Enough?:An Experiment with Data Saturation and Variability. Field Methods 2006, 18, 59-82, doi:10.1177/1525822x05279903.
- Geirhos, A.; Stephan, M.; Wehrle, M.; Mack, C.; Messner, E.M.; Schmitt, A.; Baumeister, H.; Terhorst, Y.; Sander, L.B. Standardized evaluation of the quality and persuasiveness of mobile health applications for diabetes management. Scientific Reports 2022, 12, 3639, doi:10.1038/s41598-022-07544-2.

Reviewer 2 Report
This aim of this manuscript was to assess user engagement, effectiveness, and identify barriers to adoption of a diabetic foot ulcer app developed by the team (MyFootCare). DFU is critical challenge in home care management and is expected to have worldwide impact as growing numbers are living with diabetes. The topic is timely and significant and of interest to many in the health care and home care fields.
While an important topic, the shortfall of this manuscript is its lack of scientific soundness to achieve its aims. The research design is underpowered with only 12 subjects involved in the study. The argument for only targeting 12 subjects (line 147) in the recruitment phase is weak. Substantially more subjects are needed for a serious study such as proposed. The research design should also be more intentional in terms of its target goals and how success is defined. As written, it seems quite exploratory and ill-defined.
Please check on pg 2-3 the "Error! Reference not found" citations. These may relate to the missing reference/citations for Figures 1-3 and Table 1.
The analysis of the study could also be dramatically improved. Table 2 shows declining enthusiasm for many of the features of the app (motivational image, notifications, progress graph). It would seem that tables 1 & 2 show mean data and range. It would be helpful to see SD and # of responses as well as Fig 4 shows while a drop off in utilization. Did all 12 subjects complete the study? It seems 4 subjects (P01, P04, P06, P12) did not complete using the app for 12 weeks, but also P02 stopped after week 6 (and probably should be listed under the "temporary" group. Subject P07 also seemed to discontinue usage after Week 9. Perhaps frequency count is insufficient to gauge adoption as these can be clustered early in the study. Again, with limited data, observations/results presented are anecdoctal at best. For this reviewer, it is unclear if the findings can be extended to other similar projects or merely apply to design limitations of MyFootCare. One suspects the latter which diminish enthusiasm and usefulness of this work for other similar studies. It would seem the apps' inherent design of limiting clinician involvement and data sharing is a fatal flaw that does not leverage lessons shared by telehealth researchers over the past decades.
Author Response
Dear editor,
We would like to thank you and the reviewers for your assessment of our Sensors Special Issue submission on DFU self-monitoring with the MyFootCare app. We were pleased to read the positive remarks from the editor and both reviewers, have taken their comments on board and have revised our manuscript accordingly. We believe that this has significantly improved the manuscript. Below we provide a point-by-point response to your comments and those of both reviewers and we have tracked changes to the manuscript text. Responses are separated from the comments. Manuscript text is added in italics in this point-by-point response, with revisions underlined. Line numbers are based on the manuscript without markups shown.
We hope the editor and reviewers enjoy reading this new version of our manuscript, and we look forward to their assessment.
Yours sincerely, on behalf of all authors,
Bernd Ploderer and Jaap van Netten
Reviewer 2
COMMENT 1:
This aim of this manuscript was to assess user engagement, effectiveness, and identify barriers to adoption of a diabetic foot ulcer app developed by the team (MyFootCare). DFU is critical challenge in home care management and is expected to have worldwide impact as growing numbers are living with diabetes. The topic is timely and significant and of interest to many in the health care and home care fields.
While an important topic, the shortfall of this manuscript is its lack of scientific soundness to achieve its aims. The research design is underpowered with only 12 subjects involved in the study. The argument for only targeting 12 subjects (line 147) in the recruitment phase is weak. Substantially more subjects are needed for a serious study such as proposed. The research design should also be more intentional in terms of its target goals and how success is defined. As written, it seems quite exploratory and ill-defined.
RESPONSE:
We agree with the reviewer that the topic is timely and significant. We would like to stress, however, that “effectiveness” was not included in our aims, nor has it been mentioned anywhere else in the manuscript with regards to MyFootCare. If that would have been our aim, we would have indeed needed to recruit more participants.
This perhaps explains the difference in the reviewer’s assessment of the scientific soundness of our study, and our arguments that we have performed a study using the design that is most appropriate for the aims stated. We have followed scientific consensus here, and have undertaken a mixed-methods study design to meet our aims to investigate engagement and perceived value of using MyFootCare. This is the first step in a long (and winding) scientific road towards creating an app that is effective and evidence-based. The first steps on this road are, by nature, exploratory. However, it is extremely important that these first exploratory steps are documented, and taken following a scientific approach. Rather than embarking on a quantitative journey and involving dozens or even hundreds of patients, we chose to first follow a mixed-methods approach with primarily a qualitative nature, to better understand what happens when patients start engaging with MyFootCare.
In this response, we seek to clarify that we have included an appropriate number of participants for the aims, methodology, and theoretical perspective of this work:
Firstly, we wish to clarify that effectiveness was not an aim of this manuscript. As now acknowledged in the limitations, we do not have any evidence about MyFootCare promoting DFU healing. However, this was also not our aim. As is often the case with digital health studies, the main aims were to understand user engagement (and barriers and enablers) and the perceived value for users. Understanding user engagement in a real-world context is important, because active engagement is a precondition before we can assess the efficacy or effectiveness of an app like MyFootCare. Furthermore, as mentioned in section 2.2.4, engagement with digital health interventions is more complex and varied than adherence to more traditional health interventions such as medications. It is for these reasons that starting with a more exploratory study, and publishing these findings for the scientific public to learn from, is the first step to take. And it’s that step we’re describing in this manuscript.
We revised the limitations (section 4.3) to acknowledge the lack of insight between app engagement and ulcer healing.
Lines 516-519: Due to the small cohort and the 3-month period, we did not find an association between app engagement and ulcer healing; however, our methods were also not designed to detect such an association. If we would have found one, it could also have been by chance. A larger and primarily quantitative study is needed to investigate such an association.
We also note that the lack of healing or any improvement of the ulcer was identified as a barrier in 3.3.3.
Secondly, we wish to highlight that the findings largely rest on qualitative findings. We agree that the study is underpowered for purely quantitative claims, especially for insights into the effectiveness of MyFootCare. We are therefore also not making these. However, this study relies to a large extent on qualitative data to unpack app engagement and perceived value of MyFootCare. Quantitative information is provided to give context, such as the app ratings in table 2, but the perceived value presented rests on qualitative data. Similarly, Figure 4 gives an overview of the app log data, but the enablers and barriers rely on the qualitative data. We had debates amongst the authors as to whether to present this work as a qualitative study, but we all agreed that the descriptive quantitative information is valuable, which is why we presented it as a mixed methods design. We have stated this more explicitly in the introduction.
Line 72: The specific aims of this mixed methods, but predominantly qualitative, study
Finally, the comment regarding defining goals and success may reflect differences in theoretical perspective. As noted in Section 2.2.4, this work is not based on a positivist approach (line 205), which is why the target goals and success may not appear clearly defined if observed from the quantitative viewpoint of investigating effectiveness. Instead, this work is based on an interpretivist approach, as explained in the description of the reflexive thematic analysis (section 2.2.4). The description of the analysis highlights that themes are constructed by the researchers based on their background, their interactions with the participants and app data, and the theoretical concepts they work with (as described in the bullet points in 2.2.4). Having read and reflected upon the reviewer’s comment now various times, we feel this seems be a misunderstanding of expecting an analysis of “effectiveness” in this manuscript, which would have required power calculation and a quantitative approach, while our aims required a qualitative design by nature.
To state these assumptions more clearly in the paper, we expanded the statement about the limitations of the cohort size in Section 4.3. We also sought to further justify the recruitment target of 12 participants based on empirical research on how many interviews are enough to reach data saturation [10].
Lines 156-157: Prior research also shows that data saturation often occurs after 12 interviewees [10].
Lines 512-515: This study was based on 12 participants, which limited the validity of the quantitative results presented. Whilst quantitative information is provided in figures and tables, this information is descriptive only, and the main findings about the perceived value of MyFootCare as well as barriers and enablers for engagement are based on qualitative data.
COMMENT 2:
Please check on pg 2-3 the "Error! Reference not found" citations. These may relate to the missing reference/citations for Figures 1-3 and Table 1.
RESPONSE:
Our apologies. We will convert the cross-references to plain text before submission and carefully check the PDF to avoid these errors.
COMMENT 3:
The analysis of the study could also be dramatically improved. Table 2 shows declining enthusiasm for many of the features of the app (motivational image, notifications, progress graph). It would seem that tables 1 & 2 show mean data and range. It would be helpful to see SD and # of responses as well as Fig 4 shows while a drop off in utilization.
RESPONSE:
As with comment 1, we fear that these comments are at least partly the result of this reviewer expecting a more quantitative evaluation. However, we have given all our tables and figures a critical assessment, to take this point into account.
The reviewer suggests that the standard deviation would be more helpful. However, as is described in the heading of both table 1 and table 2, numbers are median (IQR). Given that data were not normally distributed, median (IQR) are the preferred numbers to show given current scientific consensus, rather than mean (SD). To avoid further misinterpretation, we’ve changed the notation from “(x, x)” to “[x; x]”, as that is also frequently seen for the inter-quartile range. We furthermore refer to our previous answers regarding the (limited) quantitative results provided, given the predominantly qualitative nature of this study.
We politely disagree that table 2 shows declining enthusiasm for many features. We agree that the enthusiasm for notifications drops, while it decreased only slightly for motivational image; for the progress graph, the IQR is almost completely overlapping. We did not test this formally statistically, to avoid over-interpretation of these quantitative findings. Rather, we described these qualitatively. See for example the sentence immediately preceding table 2.
Line 263: and these scores decreased throughout the study
We completely agree with the reviewer 4 that Figure 4 shows a drop in utilization. This drop is seen in those with temporarily or failed engagement, and this drop is extensively discussed in section 3.3, and the discussion.
COMMENT 4:
Did all 12 subjects complete the study? It seems 4 subjects (P01, P04, P06, P12) did not complete using the app for 12 weeks, but also P02 stopped after week 6 (and probably should be listed under the "temporary" group.
Subject P07 also seemed to discontinue usage after Week 9. Perhaps frequency count is insufficient to gauge adoption as these can be clustered early in the study. Again, with limited data, observations/results presented are anecdotal at best. For this reviewer, it is unclear if the findings can be extended to other similar projects or merely apply to design limitations of MyFootCare. One suspects the latter which diminish enthusiasm and usefulness of this work for other similar studies.
RESPONSE:
We thank the reviewer for this careful assessment of Figure 4, and some spot-on observations. This is exactly why we included Figure 4, because it gives detailed insights in app usage, that go beyond the qualitative descriptions. We also agree that there is some debate possible regarding the decisions taken in the clustering. That’s exactly the debate that we’re trying to have by publishing this manuscript. Unfortunately, the vast majority of healthcare apps for diabetes never publish anything at all about their development [11], which to some extent limits drawing conclusions about whether these patterns are merely the result of the MyFootCare design, or more general. However, we give it a go in our responses below.
Firstly, regarding study completion: All 12 participants completed the study with a third and final interview at week 12. Only P02 asked to conduct his final interview earlier at week 7 due to a health concern. This has now been added to the manuscript (lines 250-252).
P02 is included in the “temporary” group, because he engaged with MyFootCare regularly for the first 6 weeks until his health deteriorated and he left the study. This was markedly different from the participants in the “failed” group, who never established a pattern of regular app use. They used MyFootCare occasionally at the start, but then stopped completely. We revised the description of the “failed group” to clarify this.
Lines 377-378: Four participants failed to engage with MyFootCare: they used the app occasionally at the start of the study but generally stopped using MyFootCare after a few weeks
We agree that the frequency count is insufficient. Whilst section 3.3 starts with the log data, the following paragraph explains that these groupings would not have been possible without the qualitative data. Furthermore, we agree that the current study is insufficient evidence and that the specific patterns and percentages are not generalizable. However, as shown in the Discussion (section 4.1, para 3), the overall trends of a majority of participants taking on average at least 1 photo per week are comparable to other studies of foot photos taken in the home. Furthermore, barriers to engagement such as environmental factors (para 4) and system-specific usability issues (para 5) also apply to other systems. We therefore think that the findings in this study are not simply and solely related to the design of MyFootCare, but useful findings reflective of barriers and limitations when creating or implementing apps for this (challenging) population.
COMMENT 5:
It would seem the apps' inherent design of limiting clinician involvement and data sharing is a fatal flaw that does not leverage lessons shared by telehealth researchers over the past decades.
RESPONSE:
We have to disagree politely with the reviewer here, both regarding the limiting of clinician involvement and data sharing, as well as this being a fatal flaw. But firstly, we completely agree with the reviewer that a telehealth system has clear merit, and we cite several successful examples where podiatrists remotely assess foot photos (lines 67-69).
However, telehealth was not the aim of MyFootCare. As we sought to describe in the introduction, MyFootCare was conceived as a tool to support the patient and the caregiver in their self-care, so that they could track and better understand their own health with the tools they already have available at home. Our key reason for this was leveraging lessons learned from psychologists and other behavioural scientists shared over the past decades. Because from this behavioural point of view, improving someone’s health starts with involving the patient in their own care. We therefore set out to create an app that would support patient in doing so.
Whilst we had concerns about clinicians monitoring patients via a telehealth system (as described in lines 548-558), we did not seek to limit clinician involvement per se. On the contrary, we were interested if MyFootCare would provide patients with data that they can share with GPs, podiatrists, and clinicians in the context of a face-to-face consultation, hoping that this may lead to patient engagement and perhaps empowerment (as described in the limitations section lines 556-558)). Also, we designed MyFootCare in such a way that these valuable lessons from telehealth learned over the past decades could be incorporated in the app. We created opportunities for data sharing and remote assessment. However, these were secondary goals, given our deliberately chosen primary objective of creating an app that starts with the patient. The findings show that most participants actually wanted to share their data with health professionals, but also that this was difficult for patients because of how busy healthcare professionals are and the limited time available during consultations.
We hope that the reviewer can assess our manuscript in light of these responses. We have purposively chosen to develop an app from a patient’s perspective, to empower them to take their more responsibility in their own footcare. We have kept options for clinician involvement, and these can be added in future versions of MyFootCare. However, for the current exploratory and predominantly qualitative assessment of app engagement and value perceptions, these did not yet have to be fully incorporated.
References
- Cassidy, B.; Reeves, N.D.; Pappachan, J.M.; Ahmad, N.; Haycocks, S.; Gillespie, D.; Yap, M.H. A cloud-based deep learning framework for remote detection of diabetic foot ulcers. IEEE Pervasive Computing 2022, 21, 78-86, doi:10.1109/MPRV.2021.3135686.
- Matilla, A.C.; Mena, A.C.; Giménez, A.A.; Fernández Morata, J. Role of the Healico© Wound Care Smartphone Application in Preventing a Foot Amputation in a 65-Year-Old Patient with Diabetes. Am J Case Rep 2022, 23, e936359, doi:10.12659/AJCR.936359.
- Boodoo, C.; Perry, A.J.; Hunter, J.P.; Duta, I.D.; Newhook, P.S.C.; Leung, G.; Cross, K. Views of Patients on Using mHealth to Monitor and Prevent Diabetic Foot Ulcers: Qualitative Study. JMIR Diabetes 2017, 2, e22, doi:10.2196/diabetes.8505.
- Anthony, C.A.; Femino, J.E.; Miller, A.C.; Polgreen, L.A.; Rojas, E.O.; Francis, S.L.; Segre, A.M.; Polgreen, P.M. Diabetic Foot Surveillance Using Mobile Phones and Automated Software Messaging, a Randomized Observational Trial. Iowa Orthop J 2020, 40, 35-42.
- Kilic, M.; KaradaÄŸ, A. Developing and Evaluating a Mobile Foot Care Application for Persons With Diabetes Mellitus: A Randomized Pilot Study. Wound management & prevention 2020, 66, 29-40, doi:10.25270/wmp.2020.10.2940.
- Swerdlow, M.; Shin, L.; D’Huyvetter, K.; Mack, W.J.; Armstrong, D.G. Initial Clinical Experience With a Simple, Home System for Early Detection and Monitoring of Diabetic Foot Ulcers: The Foot Selfie. Journal of Diabetes Science and Technology 2021, 0, 19322968211053348, doi:10.1177/19322968211053348.
- Goyal, M.; Reeves, N.D.; Rajbhandari, S.; Ahmad, N.; Wang, C.; Yap, M.H. Recognition of ischaemia and infection in diabetic foot ulcers: Dataset and techniques. Computers in Biology and Medicine 2020, 117, 103616, doi:10.1016/j.compbiomed.2020.103616.
- Jørgensen, L.B.; Sørensen, J.A.; Jemec, G.B.; Yderstræde, K.B. Methods to assess area and volume of wounds – a systematic review. International Wound Journal 2016, 13, 540-553, doi:10.1111/iwj.12472.
- Jeffcoate, W.J.; Bus, S.A.; Game, F.L.; Hinchliffe, R.J.; Price, P.E.; Schaper, N.C. Reporting standards of studies and papers on the prevention and management of foot ulcers in diabetes: required details and markers of good quality. The Lancet Diabetes & Endocrinology 2016, 4, 781-788, doi:10.1016/S2213-8587(16)30012-2.
- Guest, G.; Bunce, A.; Johnson, L. How Many Interviews Are Enough?:An Experiment with Data Saturation and Variability. Field Methods 2006, 18, 59-82, doi:10.1177/1525822x05279903.
- Geirhos, A.; Stephan, M.; Wehrle, M.; Mack, C.; Messner, E.M.; Schmitt, A.; Baumeister, H.; Terhorst, Y.; Sander, L.B. Standardized evaluation of the quality and persuasiveness of mobile health applications for diabetes management. Scientific Reports 2022, 12, 3639, doi:10.1038/s41598-022-07544-2.